# Bone Infarcts and Tumorigenesis—Is There a Connection? A Mini-Mapping Review

**DOI:** 10.3390/ijerph19159282

**Published:** 2022-07-29

**Authors:** Wojciech Konarski, Tomasz Poboży, Martyna Hordowicz, Andrzej Śliwczyński, Ireneusz Kotela, Jan Krakowiak, Andrzej Kotela

**Affiliations:** 1Department of Orthopaedic Surgery, Ciechanów Hospital, 06-400 Ciechanów, Poland; tomasz.pobozy@onet.pl; 2General Psychiatry Unit III, Dr Barbara Borzym’s Independent Public Regional Psychiatric Health Care Center, 26-600 Radom, Poland; m.hordowicz@gmail.com; 3Department of Social and Preventive Medicine, Social Medicine Institute, Medical University of Lodz, 90-647 Lodz, Poland; andrzej.sliwczynski.ahe@gmail.com (A.Ś.); jan.krakowiak@umed.lodz.pl (J.K.); 4Department of Orthopedic Surgery and Traumatology, Central Research Hospital of Ministry of Interior, Wołoska 137, 02-507 Warsaw, Poland; ikotela@op.pl; 5Faculty of Medicine, Collegium Medicum, Cardinal Stefan Wyszynski University in Warsaw, Woycickiego 1/3, 01-938 Warsaw, Poland; andrzejkotela@gmail.com

**Keywords:** secondary osteosarcoma, bone infarct, avascular necrosis, mapping review, infarct associated sarcoma, epidemiology, imaging

## Abstract

(1) Background: Avascular necrosis (AVN) may affect every part of the bone. Epiphyseal infarcts are likely to be treated early because most are symptomatic. However, meta- and diaphyseal infarcts are silent and are diagnosed incidentally. Sarcomas developing in the necrotic bone are extremely rare, but they have been reported in the literature. (2) Methods: We conducted a mapping review of recent evidence regarding these malignancies. Methods: A mapping review using a systematic search strategy was conducted to answer research questions. We limited our research to the last ten years (2012–2022). (3) Results: A total of 11 papers were identified, including 9 case reports and 3 case series. The pathomechanism of carcinogenesis in AVN was not investigated to date. Histologically, most tumors were malignant fibrous histiocytoma. The prognosis is relatively poor, especially for patients with metastases, but adjuvant chemotherapy may increase short- and long-term survival. (4) Conclusions: Since AVN-related malignancies are sporadic, no prospective studies have been conducted. The majority of evidence comes from small case series. More research is needed to identify the risk factors that would justify follow-up of patients after bone infarcts at higher risk of developing a malignancy.

## 1. Introduction

Bone infarct, also known as aseptic or avascular necrosis of the bone (AVN), is characterized by osteocytes and bone marrow element death that results from inadequate blood supply, which causes local ischemia [1]. The disease most often affects bone epiphyses but may also affect metaphyses and diaphyses. Bone epiphyses are more vulnerable to necrosis due to the lack of connection between the bone epiphyses and local blood vessels and the consequent lack of collateral circulation leads to bone ischemia. Disturbed circulation and ischemia lead to the necrosis of osteocytes and damage to the bone structure. Given that no pathogenic microorganisms are involved in this process, it is also known as sterile bone necrosis.

Sterile bone necrosis encompasses nearly forty different conditions. Nonetheless, they all are characterized by similar anatomopathological lesions and clinical courses [2]. The most common types of bone necrosis are shown in Table 1.

### 1.1. Ethological and Risk Factors for Bone Infarcts

Bone infarcts occur in both children and adults. Contributing factors may be divided into two groups, one of which are the traumatic factors. The second group includes other factors unrelated to trauma—non-traumatic factors [1,9,10].

In the first group, fractures and dislocations lead to vascularization abnormalities, eventually leading to the development of ischemia in the affected areas [11,12].

Causes of AVN unrelated to trauma include a variety of factors, such as chronic diseases, medications, and excessive alcohol consumption. The daily consumption of alcoholic beverages can cause fat deposit formation in blood vessels over time, leading to alterations in blood supply [11]. Hyperlipidemia, obesity, diabetes, and smoking were also identified to increase the risk of AVN. Many medications elevate the chances of developing osteonecrosis. Corticosteroid effects were found to be time- and dose-dependent [12,13]. Among other drugs, osteoporosis medications, especially bisphosphonates, can contribute to AVN, specifically in the jawbone. The risk is even higher in people who have received a large amount of bisphosphonates intravenously to counteract bone metastases [14,15]. Non-traumatic causes also include non-modifiable factors, such as male gender and northern and urban residence [12].

Certain medical conditions also play an essential role in the etiology of the bone infarct. These include hemoglobinopathies (e.g., thalassemia, sickle cell anemia), decompression sickness, certain autoimmune diseases, coagulopathies (antiphospholipid syndrome, protein C, and protein S deficiency), infections (HIV), and other, such as Gaucher disease and chronic liver diseases [12,16,17,18]. Medical treatments and procedures that increase the risk of AVN may include cancer treatments, radiation, dialysis, and kidney and other organ transplants [18]. Genetic polymorphisms were also found to increase the risk for this condition [17,19].

### 1.2. Bone Infarcts-Clinical Picture and Diagnosis

The primary symptom of bone infarcts is a pain in the affected bone area [12]. Initially, the pain intensifies during physical activity and disappears when it is discontinued but may be present at rest in more chronic cases. In some cases, pain may be accompanied by reduced mobility in the illness joint [20].

The patient’s history and imaging findings are usually unambiguous enough to establish an AVN diagnosis and rule out other causes of joint and bone pain. X-ray allows for diagnosis confirmation in advanced cases but may be useful in the initial differential diagnosis [12,20]. Bone scintigraphy offers the advantage of detecting abnormalities present at the earlier stages of the disease thana classic X-ray. The affected bone tissue presents with “donut-like” changes (‘cold’ in ‘hot’) [21]. Nonetheless, its use is limited because of a specificity lower than MRI. MRI has both higher sensitivity and specificity than X-ray and allows for identifying early signs of the disease [12]. Still, positron emission tomography (PET) is superior to MRI as a more sensitive imaging method, detecting early changes and allowing for the prediction of disease prognosis [10,22,23]. Its use is limited by low availability in comparison with the aforementioned methods.

### 1.3. Treatment and Management

The main aim of bone infarct treatment is the prevention of further loss of bone mass. The choice of intervention depends on the severity of the bone damage. Pharmacological treatment is mainly symptomatic [17,20]. Studies regarding nonsurgical treatment included small groups, and their quality is low—therefore, they might be regarded as experimental [17]. The medications commonly used in the treatment of AVN are listed in Table 2. 

Most patients who suffer from AVN will seek help in the advanced stages of the disease when the symptoms begin to interfere with their activities. In such patients, surgical treatment might be considered [24]. Selected surgical procedures used to treat AVN are presented in Table 3.

### 1.4. Possible Associations between Tumorigenesis and Bone Infarct

Several patient cases were published in the past linking osteonecrosis to malignant diseases. The first case of an infarct-associated sarcoma was described in 1960. This was followed by several other reports, though most describe single-patient cases or relatively small case series. Bone infarcts were portrayed as the primary cause of the tumorigenesis or secondary to it [25,26]. Infarct-related sarcomas are extremely rare even when compared with bone malignancies secondary to Paget’s disease or radiation [27,28]. Diagnosis and management are challenging, given the disease’s rarity and the paucity of available data to guide clinicians [29,30]. In addition, little is known about the pathogenesis of the disease, risk factors for the malignant transformation of the necrotic tissue, and its natural course. The last overview of the published literature, by Domson et al., was published in 2009 [30]. Therefore, we reviewed and synthesized the data on infarct-associated tumors from the last decade, intending to identify if there have been any advancements in the treatment of the disease, diagnosis, and pathogenesis, including risk factors for the development of malignancy in necrotic bone.

## 2. Materials and Methods

A mapping review of the available literature was performed. Such studies aim to screen the available literature systematically, prepare an overview of open data, and identify knowledge gaps [31]. Currently, no guidelines apply specifically to mapping reviews. We adopted a systematic search strategy to maximize the chances of identifying relevant papers. However, questions used in mapping reviews are less detailed and broader than in the case of systematic reviews, and the data is summarized narratively without grading its quality. 

Two independent reviewers screened and selected relevant publications by reviewing their abstracts and titles. The papers were then checked against the research questions. Due to the two reviewers working in parallel, we did not remove duplicates before screening to maximize the chances of identifying all relevant papers. A narrative synthesis of the results was then conducted.

### 2.1. Research Questions

What studies are available discussing a link between cancerogenesis and bone infarcts?

Which malignancies may develop in the region affected by AVN?

What is known about the pathomechanisms of tumorigenesis in necrotic bone?

What are the radiologic findings indicative of an infarct-related malignancy?

What management is preferred in these patients?

What are the outcomes of treatment and survival rates in such patients?

### 2.2. Search Strategy and Eligibility Criteria

We searched the PubMed, Google Scholar, and Cochrane databases using search phrases with keywords related to tumorigenesis, cancer, and synonyms. The results were limited to the last ten years (2012–2022). We excluded all papers in languages other than Polish, English, and Spanish. The search was performed on the 21 July 2022. Gray literature and the references of articles included in the review were also checked to identify other papers meeting the inclusion criteria.

The complete search phrases for PubMed were the following:“bone infarct*” AND (cancer OR “cancer treatment” OR “cancer patient*” OR radiotherapy OR leukemia OR neoplasm* OR carcinogenesis OR tumorigenesis OR sarcoma OR osteosarcoma)(“avascular necrosis” OR AVN) AND (cancer OR “cancer treatment” OR “cancer patient*” OR radiotherapy OR leukemia OR neoplasm* OR carcinogenesis OR tumorigenesis OR sarcoma OR osteosarcoma).

The search strategy for Google Scholar and the Cochrane database are described in the Appendix A.

The full inclusion and exclusion criteria are listed in Table 4.

## 3. Results

The search yielded a total of 233 results. Among these, eight met the inclusion criteria (Table 5). The gray literature search allowed for the identification of one other case report. A flow diagram with the reasons for the exclusion of the screened publications in Pubmed is shown in Figure 1. In the Cochrane database, 212 results met the search criteria, including 6 Cochrane reviews and 206 trials, none of which met the inclusion criteria. In the Google Search, a total of 14,615 results were found; three additional papers were identified. No new papers were found through the screening of the references.

### 3.1. Research Question 1: What Studies Discuss a Link between Cancerogenesis and Bone Infarcts?

Among the twelve studies meeting the inclusion criteria [32,33,34,35,36,37,38,39,40,41,42,43], nine were case reports [32,33,34,35,37,38,39,40,41]. Three were case series—one focused primarily on epidemiology [36], another on radiographic findings [42]. The remaining one was a case series of secondary osteosarcomas from a single institution, of which one had developed from a previous bone infarct [43]. An overview of these studies is presented in Table 5. 

**Table 5 ijerph-19-09282-t005:** Overview of studies.

First Author, Year of Publication	Type of Study	Population/ Patient Characteristics	Summary of Key Findings	Histological Type of Tumor	Location of Tumor
Alhamdan H.A., 2020 [32]	Case report	40-year-old male	A patient with sickle cell anemia was diagnosed initially with AVN. He refused to undergo THR; therefore, it was managed conservatively. Three years later, he presented with increasing pain in the left thigh. On radiographs, multiple bone infarcts were detected in both femurs, as well as in shoulders and hip joints. There were no metastases found in the chest CT and bone scintigraphy. He underwent a proximal femur resection with prosthetic reconstruction. When preparing the report, the patient was still receiving ChT.	MFH	Proximal femur
Endo M., 2012 [33]	Case report	23-year-old female	A patient with no known risk factors for AVN was followed up for 13 years using X-ray due to an infarct-like lesion in her right humerus. It was initially assessed as a benign lesion and was detected accidentally during examination for other causes. At 36-years old, the mass begun to protrude through the bone and was palpable. She underwent a joint replacement surgery and tumor resection. At a 4-year follow-up, she did not have any signs of disease progression or recurrence.	MFH	Humerus
Goel R., 2018 [34]	Case report	65-year-old female	Patient with multiple risk factors for AVN presented with a restriction in flexion in the at-risk knee. Physical examination revealed a 10 × 12 cm soft tissue mass in the lower right thigh. Bone biopsy findings and expression of MDM2 and CDK4 was indicative of low-grade osteosarcoma.	Low-grade osteosarcoma	Distal femur
Kayser G., 2017 [35]	Case report	51-year-old female	Female with a history of alcohol abuse presented with a painful mass in distal thigh. She was unable to bear weight on her lower left limb and had lost 40% of her body weight. She had anemia (7.8 g/dL of hemoglobin). Histology findings confirmed myxofibrosarcoma.	Myxfofibrosarcoma	Distal femur
Laranga R., 2022 [36]	Case series and literature review	11 patients (6 females, 5 males), with a median age of 55 years (cohort 1) and 15 cases of secondary sarcomas published between 1962–2018 (72% males, (cohort 2)	Cohort 1: 90% of patients had localized disease at the time of the diagnosis. Furthermore, 18% had grade II osteosarcoma, and the remaining 82% had grade IV. Median OS was 74 months. All patients underwent surgical treatment (including 27% with surgery alone), 64% adjuvant—ChT and 27%—neoadjuvant ChT. A total of 55% died at the end of the study. Cohort 2: 50% underwent surgery only, 43%—ChT and surgery, while 7%—palliative treatment. Survival rates at 3 years of cohort 2 and 1 were 23% and 61%, respectively. Median survival was 12 months.	Osteocarcoma	Cohort 1: Distal (64%) and proximal (18%) femur, proximal tibia (18%) Cohort 2: 50% femur, 29% tibia, 29% humerus
Lewin J., 2014 [37]	Case report	29-year-old male	Patient with bone infarct related to corticosteroid use due to Hodgins’s lymphoma in childhood. At the time of writing the paper, he was still in treatment with ChT before potential limb-sparing surgery. He will undergo evaluation and qualification for a surgical procedure after ChT.	Chondroblastic osteosarcoma	Distal femur
McDonald M.D., 2018 [38]	Case report	71-year-old male patient	Patient presented with pain in his right humerus, which increased after hearing a crack while dressing. He underwent X-ray, which showed a minimally displaced humeral fracture and a sclerotic lesion. His pain worsened over the next 2 weeks, and follow-up examination with MRI and X-ray showed a lytic lesion with periosteal reaction and a mature bone infarct. Patient did not respond well to chemotherapy and underwent an amputation. After two months, he developed metastases in lungs and lumber spine. He died 7 months after the initial presentation.	Osteogenic sarcoma	Humerus
Robbin M.R., 2013 [39]	Case report	80-year-old female	Patient presented with vague knee pain of more than 2 years’ duration. No restriction in ROM was observed. Pain was present during movements and palpation. No mass or lymphadenopathy were observed, nor were any systemic manifestations of the disease. Histology findings confirmed MFH.	MFH	Distal femur
Sivrioglu A.K., 2017 [40]	Case report	49-year-old male	Patient had a history of using steroids for the management of allergies and asthma. Multiple infarct areas were observed on X-ray and MRI, with localized changes causing cortical destruction. Histological examination confirmed a case of osteosarcoma.	Osteosarcoma (multifocal)	Distal femur
Spazzioli B., 2021 [41]	Case report	66-year-old male	The patient was diagnosed with idiopathic medullary aplasia around the age of 24, which was managed with high-dose corticosteroids and immunoglobulins and transfusions. He developed multifocal, bilateral bone necrosis in distal femurs and proximal tibias. He received 3 courses of neoadjuvant ChT and underwent femur resection with mega-prosthesis implantation. Due to periprosthetic infection, he required several reoperations.	High-grade osteoblastic osteosarcoma	Distal femur
Stacy G.S., 2015 [42]	Case series	Adult patients treated primarily at the author’s institution	From 1978 to 2008, eight patients, aged 55.1 y (31–80), with infarct-associated bone sarcomas were identified. The tibia or femoral bone were affected in all cases. In all but one patient, other sites of osteonecrosis were found. Only 50% had predisposing factors to AVN. MFH was the most common and was confirmed in 6/8 patients. The duration of pain prior to establishing a diagnosis was 4 months on average.	MFH—6/8 patients Osteosarcoma 2/8 patients	Distal femur (4/8 patients) Proximal femur (1 patient) Proximal tibia (3/8) Mid distal tibia (1 patient)
Yalcinkaya U., 2015 [43]	Case series (secondary sarcoma	Secondary sarcomas (n = 7), including 4 post-radiation, GCT related, Paget’s disease, and bone infarct.	A 59-year-old patient with bilateral infarcts of the femur and tibia and osteolytic lesion in the right distal femur. He presented with pain in the right knee. Patient had no systemic disease or other risk factors for bone infarct. The lesion was treated primarily with bone grafting and curettage. After confirmation of the malignant nature of the lesion, he was referred to the author’s institution. He underwent ChT and RxT. HE died 4 years after surgery due to lung metastasis.	Osteosarcoma	Distal femur

AVN—avascular necrosis; ChT—chemotherapy; CT—computed tomography, GTC—giant cell tumor MFH—Malignant fibrous histiocytoma; MRI—magnetic resonance; OS—overall survival; ROM—range of motion, RxT—radiotherapy.

### 3.2. Research Question 2: Which Malignancies May Develop in the Region Affected by AVN?

Different histological types of secondary sarcomas were found to arise from bone infarcts in the papers eligible for this review. These include malignant fibrous histiocytoma (MFH), fibrosarcoma, myxofibrosarcoma, and osteosarcoma [32,33,34,35,36,37,38,39,40,41,42,43]. Multifocal lesions, or the asynchronous development of malignancy in more than one bone in a single patient, were also described [42]. Nonetheless, given that most studies were short case series or single case descriptions, the list of infarct-related tumors might not be complete. 

Though it is a rare primary bone tumor, MFH was the most common type of malignant tumor secondary to bone infarction. The World Health Organization recently reclassified it as pleomorphic undifferentiated sarcoma. Immunohistological staining should be negative for S100 protein, cytokeratin, desmin, and muscle-specific actin [33,39]. Some authors suggest that such a high prevalence of MFH in association with AVN indicates that there might be a causal relation [33,39]. MFH is most prevalent in patients in the 6th decade of life and older and is twice as common in men as in women. Most cases are diagnosed at the disseminated stage, and the survival rates are relatively poor [39]. MFH was also found in 7 out of 9 patients described in the Stacy et al. case series [42]. 

Only one case of myxofibrosarcoma, which developed within a bone infarct, was described [35]. The diagnosis was confirmed with immunostaining for S-100, SOX10, and MDM2, which excluded tumors originating from the neural crest and liposarcoma [35].

Myfoxibrosarcoma arises mainly from the soft tissue and is characterized by spindle cells in a gelatinous (myxoid) background. The authors suggest that this kind of tumor, similarly to MFH, has an affinity for areas of bone affected by osteonecrosis [35]. Another case of rare, low-grade sarcoma developing in a previous bone infarct was described by Endo et al. This kind of tumor accounts for around 1% of all osteosarcomas [33].

### 3.3. Research Question 3: What Is Known about the Pathomechanisms of Tumorigenesis in Necrotic Bone?

Bone infarcts might affect the epiphysis, diaphysis, and metaphysis of long bones [40]. Epiphyseal infarcts are usually symptomatic at an early stage. The remaining two might cause pain initially but frequently are found incidentally during an imaging study ordered for some unrelated concern [34]. Given that the malignant transformation of cells is a process that requires considerable time, most secondary tumors occur in the metaphyseal and diaphyseal areas [34,40]. Tumors appear most frequently in the proximal tibia and distal femur [42]. However, one of the included papers, by Endo et al. and McDonald, described a case of the malignant transformation of infarcts in the humeral bone [33,38].

Little is known about the pathways of carcinogenesis in necrotic bone. It was previously suggested that transformation occurs due to local inflammation and reparative processes caused by the infarct [43]. When inflammatory processes become chronic, the likelihood of the malignant transformation of the reparative cells increases. Some histologic types, such as fibrosarcomas and MFH, affect areas where another lesion was already present; these include bone infarcts [35,43]. The margin of the infarct, where these processes are intensive, might be more prone to cancerous transformation. Mutations in the p53 gene were proposed to play a role by one of the authors [43]. Mutations in the p53 gene were also found in the Endo et al. MFH case [33]. The authors of other reports did not perform a molecular analysis of the cancerous tissue [34,35,36,40,41,42,43]. 

Stacy et al. stated in their manuscript that MFHs penetrate the bone cortex easier than primary tumors because they arise from the periphery of the centrally located infarct. The avascular, necrotic infarct area makes it relatively immune to tumor penetration [42]. Nonetheless, no study to date has identified pathomechanisms to support these hypotheses. Alhamdan et al. suggested that patients with sickle cell trait (SCT) (heterozygous, with fewer symptoms) might have more propensity to develop malignancies in infarcts than homozygotes (i.e., those with sickle cell disease (SCD)). The reparative processes might be more intense in the former group, thus increasing the chance of carcinogenesis [32]. Though the hypothesis is interesting, it was based solely on five cases reported in the literature to date (four with the SCT and one in a SCD patient. Furthermore, given the rarity of the disease and limited data available, it would be difficult to determine if these conditions were genuinely increasing the chances of the malignant transformation of infarcted tissue or if they develop by chance [32,37].

### 3.4. Research Question 4: What Radiologic Findings Are Indicative of Infarct-Related Malignancy?

Imaging studies are an essential step in the diagnosis of bone tumors. Infarct-associated osteosarcomas are primarily located in the lower limb, especially around the knee joint, less frequently in the proximal femur. Pain, usually around the knee or hip joints, is the leading symptom of bone infarcts and tumors [32,36,39,41,42]. X-ray is one of the primary imaging techniques used to determine the cause of pain in this location [40]. MRI and CT might also be employed in the initial diagnostic process [39]. The common finding in infarct-associated sarcomas is the presence of a well-defined sclerotic band with or without a central lucency, which corresponds to a mature bone infarct [38,39]. Nonetheless, the necrotic bone might also appear as a mixed pattern with lucencies and sclerotic changes, with no easily definable margins. Tumor radiological characteristics are described in more detail below.

#### 3.4.1. Radiographs

On X-ray, an infarcted area is visible in most cases, with a soft-tissue mass adjacent to it. Larger osteosarcomas may displace muscle shadows; on X-ray, they may present with or without aggressive features (including periosteal reaction and cortex disruption) [40,42]. In some cases, features of an upcoming fracture may also be observed. [32] Infarct-related MFHs, on the other hand, might be overlooked and diagnosed during a follow-up examination or only after a pathological fracture of the bone due to the underlying malignancy [38,42]. They tend to grow eccentrically. A soft-tissue mass might not be visible on radiographs, though discrete cortical thinning and subtle osteolysis signs might indicate MFH’s presence [42]. In Stacy et al.’s case series, only three out of seven cases could be diagnosed based on the initially taken radiographs [42]. In one case of myxofibrosarcoma, the primary finding was that extensive bone destruction by a soft-tissue mass, with periosteal reaction and some signs of mineralization in the epicenter, is indicative of a previous bone infarct [35].

#### 3.4.2. Computed Tomography

Computed tomography (CT) results are also different in the case of osteosarcomas and MFHs. In osteosarcomas, soft-tissue ossification is hyperdense to the skeletal muscle. As the mass penetrates the cortex, signs of cortical destruction are also present. Hyperdense calcifications may also be present within the tumor [32,33,42]. A hyperdense osteosclerotic band may separate osteosarcoma from the infarct [33,39]. For MFHs, a slightly hypodense appearance, in comparison to muscle tissue, is typical [42]. A hyperdense, calcified infarct margin might be present, but only residues might be visualized in some cases [33,42]. In proximity to the tumor, focal osteolysis is also a common finding. Contrast enhancement in MFH and myxofibrosarcoma was moderate; the lesion appeared slightly hyperdense in comparison to the skeletal muscle. A CT scan may also reveal the foci of newer, asynchronous bone infarcts near the tumor or in other bones [41]. For other types, no information about contrast-enhanced CTs was available [42].

#### 3.4.3. Magnetic Resonance Imaging (MRI)

Magnetic resonance is useful in imaging soft-tissue masses, interference of the tumor with surrounding blood vessels and nerves, and other adjacent structures. It enables both measurements of the tumor or the determination of multifocal lesions and the confirmation of the presence of a bone infarct. T1-weighted images of MFHs reveal a mass with mostly homogenous, intermediate, or low signal intensity (similar to cartilage or muscle tissue) [39,42]. The margin of a previous bone infarct (with apparent disruptions) was observed in all cases in the series by Stacy et al. The presence of areas of high signal intensity on T1 images, suggesting intratumor bleeding, was found in only one patient. On gadolinium-enhanced T1-weighted images, the tumor signal increased moderately in a non-homogenous manner. On T2-weighted images and fat-suppressed T2 images, MFHs were characterized by high signal intensity [42]. In addition, periosteal reaction, cortical destruction, and endosteal resorption might be seen in MFHs [38,39,42]. In the case of sarcomas, a mixed pattern of contrast enhancement is typically found (moderate to high), suggesting intra-tumor necrosis, infarct margins, and focal low-vascularity [42].

#### 3.4.4. Scintigraphy with Technetium 99-Labeled Methylene Diphosphonate

The role of scintigraphy is mostly the differentiation of bone infarcts (low radionuclide uptake) from active tumors (high uptake) and the determination of whether the disease is localized or has already disseminated [32,38,42]. Skeletal scintigraphy might also help detect multifocal lesions [35,36]. Most sarcomas secondary to bone infarct present an intense radionuclide uptake, regardless of histologic type [32,39,42]. A central portion with a decreased radiotracer uptake might be observed if the tumor arises from an area of extensive bone infarct [32,42]. Nonetheless, its use in diagnosis requires further research.

### 3.5. Research Question 5: What Management Is Preferred in These Patients?

Sarcomas are rare tumors, and those that arise from a bone infarct are even rarer. Most cases of sarcomas secondary to bone infarcts require multimodal treatment. Usually, a combination of chemotherapy (ChT) and surgical management is proposed [32,34,38,39,42]. Neoadjuvant chemotherapy might be used if limb-preserving surgery is planned [36,41]. Surgical management might include the amputation of a wide resection with endoprosthesis implantation. It may also serve as limb-salvage surgery in patients with a pathological fracture in the preoperative period. Nonetheless, such extensive resection and use of sizeable prosthetic material may provoke an infection. Such a risk must be carefully weighed against possible benefits, especially when immunocompromising ChT is administered [41].

Spazzioli et al. claimed that adjuvant and neoadjuvant ChT might increase survival rates [41]. Nonetheless, Laranga et al. have shown that the difference in 5-year overall survival between patients treated with ChT vs. surgery was not statistically significant (71% vs. 50%; *p* = 0.4773), though the survival rates were higher in that study [36].

### 3.6. Research Question 6: What Are the Outcomes of Treatment and Survival Time in Such Patients?

Osteosarcomas secondary to bone infarct carry a poor prognosis. Most patients remain symptom-free for a long period, and the diagnosis is established at stage III or IV [33,34,35,38]. The mean survival time in Laranga’s case series (cohort 1) was 74 months (95% CI, 12 not reached) for patients treated with ChT and surgery, whereas with surgery alone, it was only 20 months (95% CI, 8 not reached) [34], and it was longer than in the historical cohort (cohort 2) from the literature (12 months; *p* = 0.0247). Mortality hazard in Laranga’s cohort 1 remained lower even when adjusting for ChT administration and age when compared to the historical cohort 2, though the CI remained wide (HR: 0.333; 95% CI: 0.11–1.03). Secondary metastases occurred in 40% of patients within nine months after the initial diagnosis on average [36]. In another case series by Stacy et al., consisting of osteosarcomas and MFH, patients without metastases had a better prognosis; they lived 11 months to 25 years after the initial diagnosis. On the other hand, patients who develop metastases died (on average) within eight months after their discovery [42]. Spazzioli et al. also calculated OS for 12 cases described in the literature. The OS at 12 months and 36 months was 54% (95% CI 26–76) and 36% (95% CI 25–76), respectively. These differences might be explained by the fact that the cases included in the calculation were different than in Laranga et al.’s paper [36,41].

There is no long-term follow-up information about the single myxofibrosarcoma case reported by Kayser et al. [35]. Endo et al. have described a matter of low-grade osteosarcoma. The latter was followed up for 17 years. The patient underwent surgery 13 years after the initial diagnosis, and four years past surgery, there were no signs of disease progression [33]. Nonetheless, the authors of that paper admitted that a clinical approach consisting of regular X-rays could cause potential harm, and that low-grade osteosarcomas could be diagnosed earlier with other imaging techniques, such as CT or MRI. Therefore, there was a risk that there was a diagnostic delay in that case, and that a misdiagnosed lesion could have progressed to high-grade spindle cell sarcoma, which occurs in 1 out of 5 cases [33]. Nonetheless, more data is necessary to determine the prognosis of these tumors arising from a bone infarct.

## 4. Discussion

Ischemia of the bone is a relatively common finding in orthopedic practice. Many conditions and medications were shown to increase the risk of bone infarcts, including conditions that promote the formation of blood clots (e.g., sickle cell anemia, pregnancy, Cushing disease), steroids, alcohol abuse, diving, radiation therapy, and others [12,17,29,41]. In fact, in roughly 70% of cases, the cause of bone infarct remains unknown [30,33,40,42,43]. Not all infarcts are symptomatic, and patients remain symptom-free for a long time, which is especially true for diaphyseal and metaphyseal AVN [29,33]. That results in diagnostic and treatment delays and poorer clinical outcomes [36].

Available epidemiological studies demonstrate that less than 1% of malignant tumors affect the bone. In the USA, only 3300 cases of primary bone tumors are diagnosed each year [33]. Secondary bone tumors, including infarct-associated tumors, are even rarer, making up to 0.6–1% of all sarcomas of the bone [36,42,44]. The first cases of infarct-associated sarcomas were described by Furey et al. in 1960 [45]. In the literature, less than 150 cases of AVN-related tumors were described, inducing over 120 MFH cases [39,42]. Most patients affected were male and in their fifth or sixth decade of life [32,38,39,42]. A majority of these tumors were located in the lower limbs, especially the tibia and femur, though two case reports concerned humoral lesions [29,30,35,38]. Many cases came from small case series or single-patient case descriptions [29,30,31,32,33,34,35,36,37,38,39,40,41,42,43]. It should be underlined that silent infarcts, which are more likely to be a background for carcinogenesis and tumor development, do not present with noticeable symptoms [33]. It is possible that a growing tumor would obscure radiological signs of the past infarct. Therefore, the true epidemiology of AVN-associated tumors is likely to be underestimated [42]. On the contrary, epiphyseal bone infarcts are more likely to be discovered early, since they are usually symptomatic. Tumorigenesis, being a lengthy process, is unlikely to occur in such cases, and as a result, symptomatic infarcts are less likely to be a background for secondary malignancies [30,38,42,46].

The pathophysiology of the malignant transformation of necrotic bone tissue remains a field of hypotheses and assumptions that still need to be investigated in depth. It has been hypothesized that the reparative process and chronic inflammation lead to the transformation into sarcoma [32,33,43]. In general, the development of sarcoma in the AVN of the bones is a relatively slow process. One study of Caisson workers demonstrated that sarcoma developed 17–22 years after quitting their job. A similar timeframe applied to a case described by Endo et al., wherein the malignancy was detected 13 years after the infarct was visualized on an X-ray performed for to another reason. The close follow-up of that patient led to the detection of malignancy relatively early, before it transformed into high-grade osteosarcoma [33].

Nonetheless, the exact time interval between bone infarct and the development of bone sarcoma is unknown, since AVN timing is usually impossible to determine. In all of the cases described by Stacy et al., all patients were diagnosed when the symptoms related to the malignancy itself became apparent, and only subtle signs of previous AVN indicated that it was secondary to infarction. The situation was the same in all but one case [42]. In addition, none of the studies included in our review have proven that there is a connection between any condition, medication, or other risk factors promoting oncogenesis in patients who suffered a bone infarct [32,33,34,35,38,40]. It is yet to be established if such tumors develop by chance or if some populations need closer follow-up to allow for early detection and treatment, which could improve the long-term prognosis. Alhamdan et al. have made a suggestion that SCT patients are more likely to develop secondary malignancies in infarcted areas. Still, there is no proven connection between these two, but this concept warrants further investigation [32].

Imaging is key to diagnosis and determining signs of a previous bone infarct [33,42]. A radiologist assessing the tumor might see a well-defined, calcified band representing the previous infarct’s margins. A mixed pattern of lucencies and sclerotic changes found in the tumor’s proximity might also indicate that the disease is secondary to osteonecrosis [42]. Such changes are observed both in classic radiographs, MRIs, and CT [30,31,32,33,34,35,36,37,38,39,40,41,42,43]. Nonetheless, an X-ray is usually insufficient to assess the tumor properly, and MRI and CT are indispensable. For staging and the determination of the presence of metastases or multifocal lesions, scintigraphy and SPECT might be considered [38,41,42].

More recently, molecular markers and genomic analyses have helped determine the histological characteristics of infarct-associated tumors [33,35,38]. Immunohistochemical stains are especially useful in rare histological tumors. Myxofibrosarcomas express CD34 protein, whereas low-grade osteosarcomas express CDK4, MDM2, and mutations of the p53 protein. INI1 was retained in osteosarcoma [33,35,38]. Most of the included studies, however, did not share results of immunohistological or molecular examination.

Secondary osteosarcomas typically develop in individuals over 50 years of age, which translates to challenges in the management of the disease [36]. The overall prognosis is unfavorable in patients with sarcomas arising at the bone infarct site. Past reports have suggested that aggressive and multimodal treatment might result in 2-year survival rates comparable to primary osteosarcomas (60–70% vs. 50–80%, respectively) [30,41,42,43,47]. Unfortunately, not all patients can complete a high-dose course of ChT, due to their comorbidities, and some instead receive a shorter, low-dose regimen or none [28,36]. Some patients might not respond to such management either [38]. Therefore, overall-survival rates are much lower. In a review from 1992 by Torres and Kyriakos, only 22% of patients were alive five years after MFH diagnosis [29]. Domos et al. reported that 46% of their patients died within seven months of the diagnosis, and taking together their data and other cases published until 2004, 57% of patients were dead within 19.2 months [30].

Conversely, in the recently published case series by Stacy et al., patients who had localized disease survived 11 months to 25 years, and, in a review prepared by Laranga et al., five-year survival reached 62% (CI 28–84), and the median survival was 74 months, compared to 12 months in the cases described previously in the literature. The results were worse for patients treated with surgery only (50% lived for five years post-diagnosis and, for the case reports described previously, 50% lived for two years) [36,42]. These data suggest that surgery combined with adjuvant treatment and even surgery alone offer better results than in the past decades. However, given the small case numbers, definite conclusions cannot be drawn [29,30,36,42]. To date, there is no standardization of treatment for this type of malignancy. This might be important, especially for patients with metastases, who still have the worst prognosis, with reported survival rates at 1-year post-discovery close to 0% [29,30,41,42].

### Strengths and Limitations

Though this review mapped out recent literature regarding bone-infarct-associated malignancies, several limitations apply to our study. Mapping reviews provide a surface-level synthesis of available data and do not involve extensive and critical analysis of the main topic. This was not a full systematic review, though we applied a similar methodology in our search. We did not conduct a meta-analysis, but we were aware that this was not possible given the type of studies we expected to find. The number of analyzed cases was small, but this was also true for previously published reviews describing secondary sarcomas [30,34,36]. We have also limited the timing of publication to the last ten years because we aimed to identify any new epidemiological data and treatment outcomes changes in comparison with previous works. As a result of that restriction, fewer papers were included. On the other hand, this was the first review to focus on radiological findings and their description in a synthetic manner, as well as to discuss the hypothesized pathomechanisms of tumor development in a necrotic bone. There is a possibility that some of the papers might not have been identified, as we limited our search to one database; however, we do not think that our conclusions would change much, given the exploratory nature of this review and the fact that most papers were case reports or retrospective case series.

## 5. Conclusions

Sarcomas arising at the site of AVN are an infrequent clinical entity, with less than 200 cases described worldwide. To date, there have been no prospective studies to collect data allowing for standardization in disease management. MFH accounts for around 60% of infarct-associated tumors, though other types of osteosarcomas, including spindle-cell osteosarcoma and myxofibrosarcoma, were described. Usually, determining the presence of a previous infarct can be observed on a routine X-ray. Nonetheless, MRI/CT is essential for the assessment of the tumor. The identification of certain histological types may warrant immunostaining. To date, there is no standardization of treatment, but recent data might suggest that the overall survival improves with adjuvant ChT in localized diseases. However, overall survival is relatively low if the patient is disqualified from aggressive treatment. To date, most patients with disseminated disease did not live more than a year after the detection of metastases. More research is needed to answer questions about the pathomechanisms of the malignant transformation of bone infarcts and identify the possible risk factors that increase the chances of secondary malignancy development. That could lead to decreased mortality related to such tumors, due to early detection at a localized stage.

## Figures and Tables

**Figure 1 ijerph-19-09282-f001:**
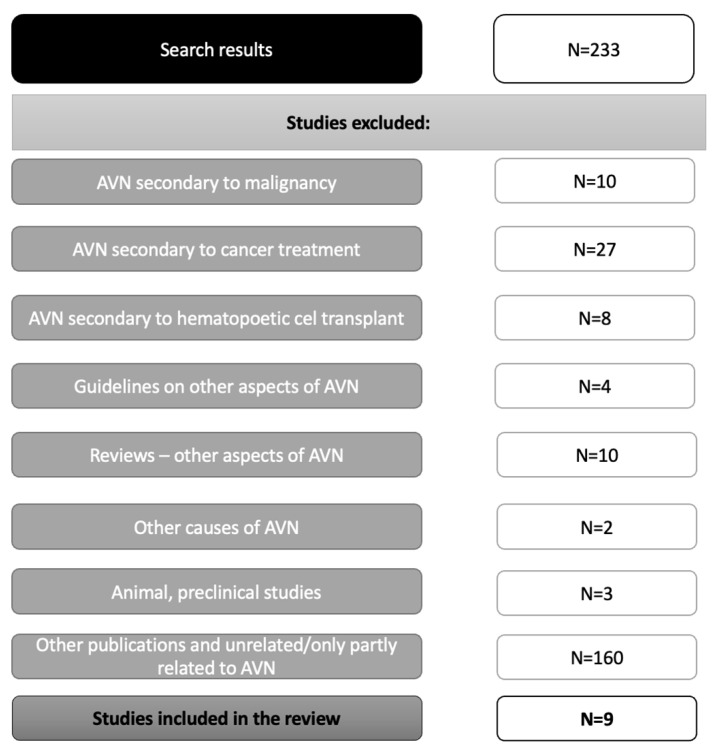
Review of the literature flow diagram of PubMed publications. AVN—Avascular necrosis.

**Table 1 ijerph-19-09282-t001:** The most common types of bone necrosis [2,3,4,5,6,7,8].

Disease Name	Bones Affected by the Disease
Scheuermann’s disease	vertebral body border plates
Haglund’s syndrome	exostosis of the posterior calcaneal tuberosity
Mueller–Weiss syndrome	tarsal navicular bone
Freiberg disease	2nd and 3rd metatarsal head
Osgood Schlatter disease	patellar tendon insertion on the anterior tibial tuberosity
Legg–Calvé–Perthes disease	femoral head

**Table 2 ijerph-19-09282-t002:** Medications used in the treatment of AVN [12,17,20].

Classes of Drugs	Examples	Role in Osteonecrosis Management
Non-steroidal anti-inflammatory drugs	Ibuprofen or naproxen	Help relieve pain and inflammation associated with AVN
Osteoporosis medications	Alendronic acid	Some studies indicate that osteoporosis drugs can slow the progression of AVN.
Hypolipidemic drugs	Statins, fibrates	Prevention of micro and macro angiopathies
Anticoagulants and antiplatelet agents	Warfarin, acetylsalicylic acid	Inhibition of thrombus formation and anti-aggregation effects

**Table 3 ijerph-19-09282-t003:** Surgical procedures used to treat AVN [9,12,17,19,24].

Treatment	Characteristics
Spinal decompression	During this procedure, the surgeon removes part of the inner layer of bone. In addition to reducing pain, this treatment has the effect of stimulating osteogenesis and neovascularization.
Bone graft (transplant)	The procedure helps to strengthen the area of bone affected by the lesions. During the procedure, some healthy bone taken from another part of the body is used.
Bone osteotomy	During this procedure, a bone wedge above or below the stressed joint is removed—This helps to shift weight away from the damaged bone. Changing the shape of the bone may allow the joint replacement surgery to be pushed back.
Joint replacement (alloplasty)	This treatment is used when other treatments do not help; it involves replacing the damaged parts of the joint with plastic or metal parts.

**Table 4 ijerph-19-09282-t004:** Inclusion and exclusion criteria.

Inclusion Criteria	Exclusion Criteria
Papers published in 2012–2022 Language: English, Polish, Spanish Papers discussing patient cases of malignant disease secondary to bone infarct/AVN Papers describing pathophysiological and clinical aspects (e.g., radiological findings, diagnosis, management) of bone neoplasms developing in the infarcted areas	Papers published before 2012 Language other than specified in inclusion criteria Papers describing patient cases of bone malignancy secondary to a disease other than AVN (such as bone metastases, Paget’s disease of the bone, chronic osteomyelitis, fibrous dysplasia), medications (including anticancer treatment), or other factors (e.g., radiation) Papers describing different aspects of bone changes secondary to clinical entities other than bone infarct/AVN Animal and preclinical studies

## Data Availability

Additional data are available from the corresponding author upon reasonable request.

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
