# Peer review of "Bone Infarcts and Tumorigenesis—Is There a Connection? A Mini-Mapping Review"

_ijerph, 2022, doi:10.3390/ijerph19159282_

Round 1
Reviewer 1 Report
The manuscript titled, "Bone infarcts and tumorigenesis – is there a connection? A mini-mapping review", reports a mapping review of recent evidence regarding avascular necrosis (AVN) that may affect every part of the bone. According to the review, epiphyseal infarcts are likely to be treated early because most are symptomatic. However, meta- and diaphyseal infarcts are silent and are diagnosed incidentally. The manuscript is well written with all relevant citations and it's very informative.
Author Response
Dear Reviewer,
We are thankful for your opinion about our manuscript. We hope that it will meet with similar appreciation from other readers.
Best regards,
Wojciech Konarski
Reviewer 2 Report
This study by Konarski et al., is a review of the literature investigating the relatively rare bone-infarct associated malignancies. The authors collected case reports published in a 10-year period to identify potential pathological mechanisms associated with avascular necrosis and bone tumorigenesis. They discuss potential risk factors, diagnosis and treatment methods used to identify bone infarcts associated tumors. The authors conduct a thorough analysis of the case reports, and present the radiological and other imaging related findings in a detailed manner. This is a good study that is timely and sheds light on a rare disease.
Author Response
Dear Reviewer ,
Thank you for your opinion. We truly appreciate your positive review. We understand that the quality of evidence is not high, but we believe that this would motivate other researchers to study secondary bone tumors in more depth.
Best regards,
Wojciech Konarski
Reviewer 3 Report
Konarski et al. performed a mapping review about bone infarcts and tumorigenesis. The paper is well written, and the topic is of importance. However, here I raise a question since the exclusion criteria are not completely clear. For example, why do authors decide to exclude cases of bone malignancy secondary to disease other than AVN, and how do they define those situations? For instance, Alhamdan et al. (https://doi.org/10.1016/j.ijscr.2020.11.004) show a cancer case related to bone infarct in a sickle cell patient, but no clear relation of cancer to sickle cell anomaly itself was provided. Why was this paper excluded? Also, why were other cancer cases removed from the analysis, as Lewin et al. l (https://doi.org/10.1089/jayao.2014.0021)?
Author Response
Dear Reviewer, this is an excellent observation. We wanted to exclude cases of bone tumors secondary to another disease than bone infarcts/avascular necrosis (which, as stated in the first paragraph in the introduction section, are used synonymously). Metastatic bone tumors (such as in breast or prostate cancers), bone changes secondary to some hematological diseases (such as multiple myeloma, or myeloproliferative neoplasms), were excluded. Also, osteosarcomas developing on the basis of other diseases, such as Paget’s disease, chronic osteomyelitis, or fibrous dysplasia (see: DOI: 10.1016/j.prp.2017.10.018) were not suitable for the analysis. We have included more examples of such exclusions in table 4 with selection criteria, to make our description of the selection criteria more understandable.
For instance, Alhamdan et al. (https://doi.org/10.1016/j.ijscr.2020.11.004) show a cancer case related to bone infarct in a sickle cell patient, but no clear relation of cancer to sickle cell anomaly itself was provided. Why was this paper excluded? Also, why were other cancer cases removed from the analysis, as Lewin et al. l (https://doi.org/10.1089/jayao.2014.0021)?
Due to the screening nature of mapping reviews (which are not equivalent to a full systematic review), it is possible that some papers were not identified through our search, especially when we limited it to one database. This is true for Alhamdan et al. paper, where other keywords were used. However, after modifying the initial search phrase by adding “osteosarcoma” and “sarcoma”, that paper showed up in the search results (only 6 more papers were identified, the other one being McDonald et al. 2018 – already included). The other paper by Lewin et al. was not indexed in the PUBMed’s database, which explains why it was not included. Nonetheless, another reviewer has suggested we search through the Google Scholar and Cochrane databases. With that search, we have identified additional 3 papers meeting the inclusion criteria, including Alhamdan et al, Lewin et al, and Spazzioli’s et al. We added them to the analysis. All of these papers reported single cases of infarct-associated sarcomas.
Best regards,
Wojciech Konarski
Reviewer 4 Report
In this study, Konarski et al described the role of AVN-related malignancies. The malignant transformation of AVN is exceedingly rare. The authors tried to link AVN with malignant transformation. Due to limited data in the literature, there is no clear molecular mechanism connecting AVN with malignant transformation. My comments are:
1: Only PubMed is search is done. Include google scholar and Cochrane database search. In this study, only 8 studies were included of which 6 papers are case reports with meager scientific yield.
2: Table 5: no need for country of study. Check how these patients were followed and any specific chemotherapeutic regimens received.
3: in lines 203-223 the pathophysiology of AVN and malignant transformation is vaguely described. Describe the molecular mechanisms of AVN.
For a better understanding of malignant transformation, it is prudent to understand the molecular mechanism of AVN. Though very little to no data is available to describe the molecular mechanism to describe the malignant transformation of AVN. However, there are studies that explain the molecular mechanism of AVN. Explaining these mechanisms might be helpful for future researchers to focus on better understanding the pathophysiology of the malignant transformation of AVN.
PMID: 26213697
4: The last paragraph should be a personalized therapeutic approach based on molecular alterations of these rare tumors.
Author Response
Dear Reviewer. We are thankful for your opinion about our manuscript.
AD 1
We added these two databases to our search. Additional 3 references were identified through Google Scholar Search. In the Cochrane Library, a total of 238 publications were identified, including 6 reviews and 206 trials. None of the publications found met the inclusion criteria. We identified new case reports identified through Google Scholar. We included the details of the Cochrane and Google Scholar search phrases in supplementary file 1.
AD 2
There were no such descriptions, and long-term follow-up is missing for most patients. Aggregated data on follow-up for up to 5 years were available from Laranga et al. and Spazzioli et all papers, and these were included in our paper.
AD 3
We agree that AVN pathophysiological mechanisms are interesting. These were described extensively in other papers. AVN’s molecular mechanisms are out of the scope of our mapping review. It is also important to highlight that mapping reviews are to detect trends and advancements in a given research field, not to synthesize all available evidence (as in systematic reviews). In addition, our mapping review was meant to describe secondary malignancies, not the primary cause, though basic concepts of this heterogenous clinical entity were given in the introduction.
AD 4
We would like to be able to make such recommendations. That was also one of the aims of our manuscript. After reviewing available evidence (including 3 additional studies), we came to the conclusion that the current state of evidence does not allow for it. Only the paper by Endo et al. includes molecular diagnosis. We would like to highlight that that was also not the aim of our mapping review, which was specified in the questions described in “methods”.
Best regards,
Wojciech Konarski